

# On determining the Point of no Return in Climate Change

Brenda C. van Zalinge[1], Qing Yi Feng[1], and Henk A. Dijkstra[1]

[1]Institute for Marine and Atmospheric Research Utrecht, Utrecht University, Utrecht, the Netherlands

*Correspondence to:* Qing Yi Feng (Q.Feng@uu.nl)

**Abstract.** Earth's Global Mean Surface Temperature (GMST) has increased by about 1.0°C over the period 1880-2015. One of the main causes is thought to be the increase in atmospheric greenhouse gases (GHGs). If GHG emissions are not substantially decreased, several studies indicate there will be a dangerous anthropogenic interference (DAI) with climate by the end of this century. However,

there is no good quantitative measure to determine when it is "too late" to start reducing GHGs in order to avoid DAI. In this study, we develop a method for determining a so-called Point of No Return (PNR) for several GHG emission scenario's. The method is based on a combination of stochastic viability theory and uses linear response theory to estimate the probability density function of the GMST. The innovative element in this approach is the applicability to high-dimensional climate

models as is demonstrated by results obtained with the PLASIM climate model.

## 1   Introduction

In the year 2100, which is as far away (or as close) as 1932 in the past, mankind will be living on an Earth with, at least, a different climate than today. At that time, we will know the 2100-mean Global Mean Surface Temperature (GMST) value and its increase, say $\Delta T$, above the pre-industrial GMST

value. From the GMST records of the $21^{nd}$ century, it will then also be known whether this change in GMST has been gradual or whether it was rather 'bumpy'. If the observational effort will continue as of today, there will be also an adequate observational record to determine whether the probability of extreme events (e.g., flooding, heat waves) has increased.

The outcomes of these future observations, to be made by our children and/or grandchildren, will

strongly depend on socio-economic and technological developments and political decisions which are made now and over the next decades. Fortunately, there is a set of rational tools available to inform decision makers: Earth System Models. These models come in different flavours, from global climate models (GCMs) aimed to provide details on the development of the ocean-atmosphere-ice-land system to integrated assessment models (AIMs) which also aim to describe the development of

the broader socio-economic system. During the preparation for the fifth assessment report (AR5) of the Intergovernmental Panel of Climate Change (IPCC), GCM studies have focussed on the climate



system response to GHG changes as derived by AIMs from different socio-economic scenario's; the data is gathered in the so-called CMIP5 archive (Pachauri et al., 2014).

Depending on the representation of fast climate feedbacks in GCMs, determining their climate
sensitivity, the CMIP5 models project a GMST increase of 2.5-4.5°C over the period 2000-2100. This does not mean that the actual measured value of $\Delta T$ in 2100 will be in this interval. For example, the GMST may be well outside because of current model errors which misrepresented the strength of a specific feedback. As a consequence, a transition occurred in the real climate system during the period 2016-2100, which did not occur in any of the CMIP5 model simulations. Another
possibility is that the GHG change eventually was far outside of the scenario's considered in CMIP5.

A crucial issue in 2100 will be whether a climate state has been reached where a dangerous anthropogenic interference (DAI) can be identified (Mann, 2009). In this case, present-day islands will have been swallowed by the ocean, extreme events have increased in frequency and magnitude as envisioned in the Burning Embers diagram (Smith and Schneider, 2009). These effects are then
very inhomogeneously distributed over the Earth and have lead to enormous socio-economic consequences. If this is the case in 2100, then there is a point in time where we must have crossed the conditions for DAI. This time, marking the boundary of a 'safe' and 'unsafe' climate state, obviously depends on the metrics used to quantify the state of the complex climate system.

In very simplified views, this boundary is interpreted as a threshold on pCO$_2$ (Hansen et al., 2008)
or on GMST. The latter, in particular the $\Delta T_c = 2°C$ threshold, has become an easy to communicate (and therefore leading) idea to set mitigation targets for CO$_2$ reduction. Emission scenario's have been calculated (Rogelj et al., 2011) such that $\Delta T$ will remain below $\Delta T_c$. Although thresholds on GMST have been criticized for being very inadequate regarding impacts (Victor and Kennel, 2014), such a threshold (the aim is $\Delta T_c = 1.5°C$) forms the basis of policy making as is set forward in the
Paris 2015 (COP21) agreement.

Suppose that measures are being taken to keep $\Delta T < \Delta T_c$, does this mean that we are 'safe'? The answer is a simple no, as regionally still DAI may have occurred such as the disappearance of island chains due to sea level rise (Victor and Kennel, 2014). Hence, attempts have been made to define what 'safe' means in a more general way, such the Tolerable Windows Approach (TWA)
(Petschel-Held et al., 1999) and Viability Theory (VT) (Aubin, 2009). These approaches also deal with general control strategies to steer a system towards 'safety' when needed.

On a more abstract level, both TWA and VT start by defining a desirable (or 'safe') subspace $V$ of a state vector $\mathbf{x}$ is a general state space $X$. This subspace is characterized by constraints, such as thresholds on properties of $\mathbf{x}$. An example is the threshold $\Delta T < \Delta T_c$ on GMST, where $\mathbf{x}$ is in
general a high-dimensional state vector in a GCM. When the time-development (or trajectory) of $\mathbf{x}$ is such that it moves outside the subspace $V$, a control is sought to steer the trajectory back into $V$. Note that this is an abstract formulation of the mitigation problem, when the amplitude of the emission of greenhouse gases is taken as control. Recently, Heitzig et al. (2016) have added more



detail to regions in the space $X$ which differ in their 'safety' properties and amount of flexibility in control to steer to 'safety'.

Giving a certain desirable subspace of the climate system's state vector (i.e., to avoid DIA) and a suite of control options, (i.e., $CO_2$ reduction) it is important to know when it is too late to be able to steer the system to 'safe' conditions, say at the year 2100. In other words, when is the Point of No Return (PNR)? The TWA and VT approaches, and the theory in Heitzig et al. (2016), suffer from the 'curse of dimensionality' and cannot be used within CMIP5 climate models. For example, the optimization problems in VT and TWA lead to dynamic programming problems which have only been solved for model systems with low-dimensional state vectors. The approach in Heitzig et al. (2016) requires the computation of region boundaries in state space, which also becomes tedious in more than two dimensions. Hence, with these approaches it will be impossible to determine a PNR using reasonably detailed models of the climate system.

In this paper, we present an approach similar to TWA and VT, but one which can be applied to high-dimensional models of the climate system. Key in the approach is the estimation of the probability density function of the properties of the state vector $\mathbf{x}$ which determine the 'safe' subspace $V$. The PNR problem is also coupled to limitations in the control options and can be defined precisely within our approach. The more abstract formulation is presented in section 2 below, building on stochastic viability theory and control strategies. Just to illustrate the concepts, we apply the approach in section 3 to an idealized energy balance model with and without critical conditions (bifurcation behavior). In section 4, the application to a high-dimensional climate model follows, using data from the Planet Simulator (PLASIM, (Fraedrich et al., 2005)). We combine these results with control strategies on GHG emissions and the associated cost functions (Stern, 2007) to find the PNR and optimal mitigation scenario for different GHG emission (RCP) scenario's using the PLASIM model. A summary and discussion in section 5 concludes the paper.

## 2   Methodology

In this section we briefly describe the concepts we need from stochastic viability theory and then define the PNR specifically for the climate change problem.

### 2.1   Stochastic viability theory

Viability theory basically studies the control of the evolution of dynamical systems to stay within certain constraints on the system's state vector (Aubin, 2009). These constraints define a viable region $V$ in state space. For a finite dimensional deterministic system, with state vector $\mathbf{x} \in R^d$ and vector field $\mathbf{f} : R^d \to R^d$, given by

$$\frac{d\mathbf{x}}{dt} = \mathbf{f}(\mathbf{x}, t), \tag{1}$$



an initial condition $\mathbf{x}_0 = \mathbf{x}(t=0)$ is called viable if $\mathbf{x}(t) \in V$, for all $0 \le t \le t^*$, where $t^*$ is a certain end time. The set of all these initial conditions forms the viability kernel associated with $V$. In a more general formulation of viability theory, an input is also considered in right hand side of Eq. (1) which can be used to control the path of the trajectory $\mathbf{x}(t)$ in state space.

Stochastic extensions of viability theory consider finite dynamical systems defined by stochastic differential equations

$$d\mathbf{X_t} = \mathbf{f}(\mathbf{X}_t, t)dt + \mathbf{g}(\mathbf{X}_t, t)d\mathbf{W}_t, \tag{2}$$

where $\mathbf{X}_t$ is a multidimensional stochastic process, $\mathbf{W}_t$ the standard multidimensional Wiener process and the vector field $\mathbf{g}$ describes the dependence of the noise on the state vector. The normalised probability density function (PDF) $p(\mathbf{x}, t)$ can be formally determined from the Fokker-Planck equation associated with Eq. (2).

A stochastic viability kernel $V_\beta$ consists of initial conditions $\mathbf{X}_0$ for which the system has, for $0 \le t \le t^*$, a probability larger than a value $\beta$ to stay in the viable region $V$ (Doyen and De Lara, 2010). For example, in a one-dimensional version of Eq. (2) with state vector $X_t \in R$ and with a viable region $V$ given by $x \le x^*$ a state $X_t$ is called viable, with tolerance probability $\beta_T$, if

$$\int_{-\infty}^{x^*} p(x, t) \, dx \ge \beta_T, \tag{3}$$

and otherwise, $X_t$ is called non viable.

## 2.2 Point of no return

In the climate change problem, scenario's of GHG increase and the associated radiative forcing have been formulated as Representative Concentration Pathways (RCPs). In Pachauri et al. (2014), there are four RCP scenarios (Fig. 1a) ranging from an increase in radiative forcing of $2.6 \ \mathrm{Wm}^{-2}$ (RCP2.6) at 2100 (with respect to 2000) to an increased forcing of $8.5 \ \mathrm{Wm}^{-2}$ (RCP8.5).

To define the PNR for each of these RCPs, a collection of mitigation scenario's is described by functions $F_\lambda(t)$, where $\lambda$ is a parameter. For instance, the collection $F_\lambda$ could result from mitigation measures that lead to an exponential decay to different stabilisation levels (measured in $CO_2$ equivalent, or $CO_2$eq) within a certain time. An example of such a collection $F_\lambda$ is shown by the dashed and dotted red lines in Fig. 1b. The most extreme member of $F_\lambda$ is defined as the mitigation scenario (represented by a certain value of parameter $\lambda$) which has the steepest initial decrease at a certain time $t$ (dashed curve in Fig. 1b).

When forcing a climate model with a RCP scenario, its state vector $\mathbf{X}_t$ may not be viable anymore at a certain time. The first year of non viability of $\mathbf{X}_t$ is indicated by $t_b$ (Fig. 1b). Once $\mathbf{X}_t$ is not viable, ideally we want to control the $CO_2$eq concentration directly such that $\mathbf{X}_t$ will become viable again. However, reducing $CO_2$eq emissions is accompanied with technological, social, economic




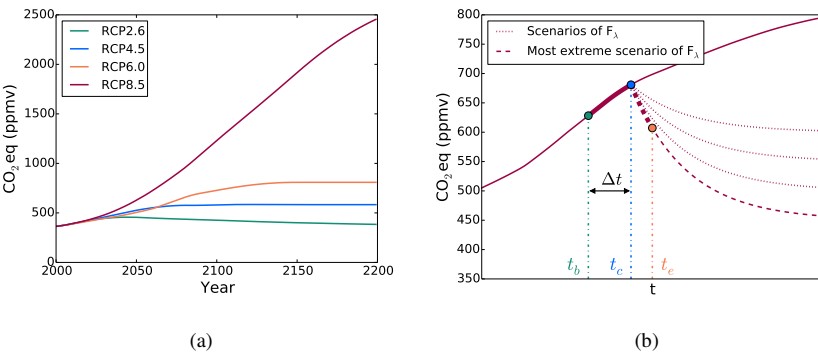

(a)                                                    (b)

**Figure 1.** *(a) $CO_2eq$ trajectories of the RCP scenario's used by the IPCC in CMIP5. (b) The solid red curve represents a typical RCP scenario. At the time $t_b$ the climate state becomes non-viable, while at time $t_c$ a mitigation scenario $F_\lambda$ is applied; at time $t_e$, the climate state is viable again.*

and institutional challenge, and therefore mitigation will be delayed $\Delta t$ years after the first year of non viability (Pachauri et al., 2014). The moment at which the $CO_2eq$ mitigation scenario is applied, is the time of action $t_c$; $t_c = t_b + \Delta t$. From the time of action, the $CO_2eq$ concentration will decrease according to one of the mitigation scenarios in $F_\lambda$. Eventually, $\mathbf{X}_t$ may become viable again and this point in time is indicated by $t_e$ (Fig. 1b).

For a given RCP scenario, tolerance probability $\beta_T$, viable region $V$ and collection $F_\lambda$, we define the PNR ($\pi_t$) as the first year where, even when at that moment the most extreme mitigation scenario of $F_\lambda$ is applied,

(a) either $\mathbf{X}_t$ will be non viable for more than $\tau_T$ years, where $\tau_T$ is a set tolerance time, or

(b) $\mathbf{X}_t$ will be non viable in 2100.

The first PNR, which we will indicate below by $\pi_t^{tol}$, is based on limiting the amount of years that $\mathbf{X}_t$ is non viable, since during these years society is exposed to risks from, for example, extreme weather events. The second PNR, which we will indicate below by $\pi_t^{2100}$ imposes no restrictions on how long $\mathbf{X}_t$ is non viable, but it only requires that $\mathbf{X}_t$ is non viable at the end on the century. We will use both PNR concepts in the results below.

## 3 Energy balance model

In this section, we illustrate the concepts and the computation of the PNR for an idealized energy balance model of Budyko-Seller type (Budyko, 1969; Sellers, 1969).



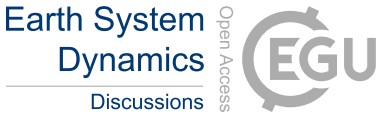

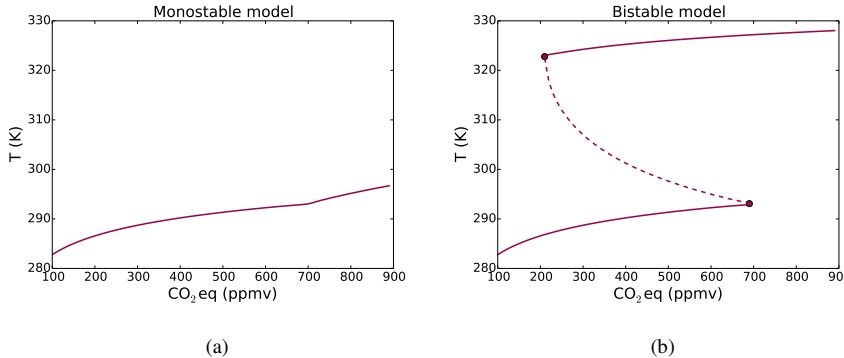

(a)                  (b)

**Figure 2.** *Bifurcation diagram for $\alpha_1 = 0.45$ ((a), monostable model) and $\alpha_1 = 0.2$ ((b), bistable model). The solid line represents a stable equilibrium while a dashed line represents an unstable equilibrium.*

### 3.1 Formulation

We use the stochastic extension of the model formulation as in Hogg (2008). The equation for the atmospheric temperature $T_t$ (in K) is given by

$$dT_t = \frac{1}{c_T}\Big\{Q_0(1-\alpha(T_t)) + G + A\ln\frac{C(t)}{C_{ref}} - \sigma\epsilon T_t^4\Big\}dt + \sigma_s dW_t. \tag{4}$$

The values and meaning of the parameters in Eq. (4) are given in Table 1. The first term in the right hand side of Eq. (4) represents the short-wave radiation received by the surface and $\alpha(T)$ is the albedo function, given by

$$\alpha(T) = \alpha_0 H(T_0 - T) + \alpha_1 H(T - T_1) + \big(\alpha_0 + (\alpha_1 - \alpha_0)\frac{T - T_0}{T_1 - T_0}\big)H(T - T_0)H(T_1 - T). \tag{5}$$

This equation contains the effect of land ice on the albedo and $H(x) = (1 + \tanh(x/\epsilon_H))/2$ is a continuous approximation of the Heaviside function. When the temperature $T < T_0$, the albedo will be $\alpha_0$ and when $T > T_1$ it will be $\alpha_1$ and the albedo is linear in $T$ for $T \in [T_0, T_1]$. The second term in the right hand side of Eq. (4) represents the effect of greenhouse gases on the temperature. It consists of a constant part ($G$), and a part ($A\ln\frac{C(t)}{C_0}$) depending on the mean $CO_2$eq concentration in the atmosphere (indicated by $C(t)$). The third term in the right hand side of Eq. (4) expresses the effect of long-wave radiation on the temperature and the last term represents noise with a constant standard deviation $\sigma_s$. The standard value of $\sigma_s$ is such that the variance of the noise ($\sigma_s^2$) is 3% of the value of $G/c_T$. The variance in $CO_2$ concentration originates mostly from seasonal variations, but the 3% is on the high side. Nevertheless, we still use this value, because if we take values smaller than 3% the PDF of the GMST will almost be a delta function and concepts can not be illustrated clearly.





**Table 1.** *Value and meaning of the parameters in the energy balance model given by Eq. (4).*

| | | | | | |
|---|---|---|---|---|---|
| $c_T$ | $5.0 \times 10^8$ Jm$^{-2}$K$^{-1}$ | Thermal inertia | $\epsilon$ | 1.0 | Emissivity |
| $Q_0$ | $342$ Wm$^{-2}$ | Solar constant/4 | $\alpha_0$ | 0.7 | Albedo parameter |
| $G$ | $1.5 \times 10^2$ Wm$^{-2}$ | Constant | $\alpha_1$ | 0.2 or 0.45 | Albedo parameter |
| $A$ | $2.05 \times 10^1$ Wm$^{-2}$ | Constant | $T_0$ | 263 K | Albedo parameter |
| $C_{ref}$ | 280 ppmv | Reference $CO_2$ concentration | $T_1$ | 293 K | Albedo parameter |
| $\sigma$ | 5.67 x $10^{-8}$ Wm$^{-2}$K$^{-4}$ | Stefan Boltzmann constant | $\epsilon_H$ | 0.273 K | Albedo parameter |

### 3.2   Results: stochastic viability kernels

When using the global mean $CO_2$eq concentration $C$ in Eq. (4) as a time-independent control param-
eter, a bifurcation diagram can be easily (numerically) calculated for the deterministic case ($\sigma_s = 0$).
In Fig. 2, such diagrams are plotted of $C$ versus the equilibrium temperature $T$ for two values of
$\alpha_1$. To obtain realistic values for the temperature, the equilibrium temperature equilibria are shifted
upwards by 30 K. This is done by substituting $T$ with $T - 30$ and adapting the right hand side of
Eq. (4) such that the new temperature is a steady state. The diagram corresponding to $\alpha_1 = 0.2$ (Fig.
2a) has two saddle-node bifurcations which are absent for $\alpha_1 = 0.45$ (Fig. 2b). From now on, the
energy balance model with $\alpha_1 = 0.45$ and $\alpha_1 = 0.2$ will be called monostable case and bistable case,
respectively.

For $\sigma_s \neq 0$, we explicitly determine the normalised PDF $p(x,t)$. Rewriting Eq. (4) as

$$dT_t = f(T_t, t)dt + \sigma_s dW_t, \tag{6}$$

with $f(T,t) = c_T^{-1}(Q_0(1 - \alpha(T)) + G + A\ln\frac{C(t)}{C_{ref}} - \sigma\epsilon T^4)$, the Fokker-Planck equation of Eq. (6)
is given by

$$\frac{\partial p}{\partial t} + \frac{\partial(fp)}{\partial x} - \frac{\sigma_s^2}{2}\frac{\partial^2 p}{\partial x^2} = 0. \tag{7}$$

This differential equation is solved numerically for $p(x,t)$ under any prescribed function $C(t)$ with
boundary conditions $p(x_u,t) = p(x_l,t) = 0$, where $x_l = 270$ K and $x_u = 335$ K, and an initial con-
dition $p(x,0)$ (specified below) satisfying $\int_{x_l}^{x_u} p(x,0)dx = 1$.

We first show stochastic viability kernels for each initial condition $T_0$ and $C_0$, where $C_0$ is an
initial $CO_2$eq concentration and $T_0$ is the expectation value of the initial PDF of $T_t$. As starting time,
we take the year 2030 and suppose that the climate system will be forced by a certain RCP scenario
from 2030 till 2200. For every $C_0$, the original RCP scenario from Fig. 1a is adjusted such that its
time development remains the same, but it has $C_0$ as $CO_2$eq concentration in 2030. The PDF of the
GMST $p(x,t = 0)$ ($t = 0$ refers to the year 2030) has a prescribed variance (defined by $\sigma_s^2$) and an
expectation value $T_0$.

In Fig. 3, the stochastic viability kernels are plotted for the energy balance model forced by the
RCP4.5 scenario and a viable region $V$ defined by $T \leq 293$ K. The results for the monostable and



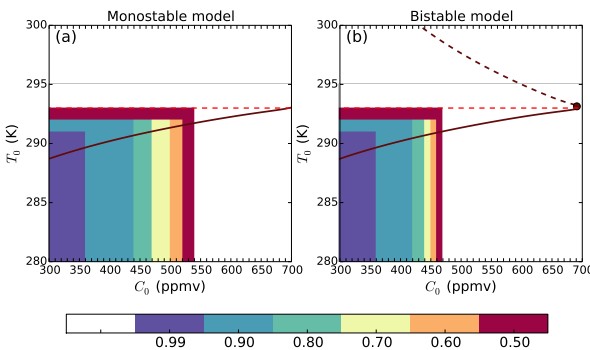

**Figure 3.** The stochastic viability kernels for the monostable and bistable cases forced by the RCP4.5 scenario. The viable region is defined as $T \leq 293K$ and is indicated by the red dashed line. This plots show, for each combination of $T_0$ and $C_0$, in which stochastic viability kernel these initial values are located. The numbers in the colorbar stand for the $\beta$ in $V_\beta$. For convenience, the bifurcation diagram of the deterministic model is also shown.

bistable cases are plotted in Fig. 3a and Fig. 3b, respectively. The colors indicate for each combination of $T_0$ and $C_0$ in which stochastic viability kernel the initial state $(C_0, T_0)$ is located. For example, consider the bistable case and an initial condition of $T_0 = 288$ K and $C_0 = 400$ ppmv, then this initial condition is in the kernel $V_\beta$ with $\beta \geq 0.9$. The white areas contain initial conditions that are in a stochastic viability kernel $V_\beta$ with $\beta < 0.5$.

The sensitivity of the stochastic viability kernels with respect to RCP scenario, threshold defining the viable region $V$ and amplitude of the noise $\sigma_s$ was investigated (results not shown). The behaviour is as one can expect in that the area of the kernels becomes smaller (larger) when noise is larger (smaller), the threshold temperature is smaller (larger) and when the radiative forcing associated with the RCP scenario is more (less) severe. For example for the RCP6.0 scenario, each
combination of $T_0$ and $C_0$ (same range as in Fig. 3) is in a $V_\beta$ with $\beta < 0.5$ for both mono- and bistable cases.

### 3.3 Results: Point of No Return

We choose the collection $F_\lambda$ to consist of mitigation scenarios that exponentially decay to the preindustrial $CO_2$eq concentration, which is 280 ppmv. For this exponential decay, we consider different
e-folding times, which is indicated by the parameter $\tau_d$. As most extreme mitigation scenario an exponential decay within 50 years is considered, which corresponds to an e-folding time of $\tau_d = 9$ years. Finally, the collection $F_\lambda$ consist of mitigation scenarios for which $\tau_d \geq 9$ years. A mitigation scenario with an e-folding time of $\tau_d$ years is then given by

$$F_\lambda(t) = (C_{t_c} - 280) \exp\left(-\frac{t - t_c}{\tau_d}\right) + 280. \tag{8}$$




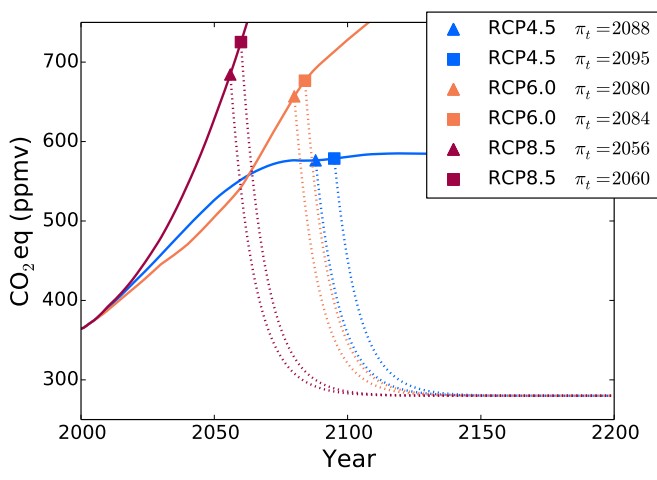

(a)

**Figure 4.** The PNR $\pi_t^{tol}$ for a system forced with different RCP scenarios, tolerance probability $\beta_T = 0.9$ and tolerance time $\tau_T = 20$ years. The triangles indicate the point of no return for the bistable case and the squares for the monostable case. The dotted line is the most extreme scenario of $F_\lambda$ with an exponential decay to 280 ppmv and an e-folding time of 9 years. Note that for both cases there is no PNR when the model is forced with the RCP2.6 scenario.

In this equation, $t_c$ is the time at which the mitigation scenario is applied and $C_{t_c}$ the associated $CO_2$eq concentration at that moment.

     Next, we determine PNR values $\pi_t^{tol}$ for the energy balance model when it is forced by the four different RCP scenarios using a tolerance probability of $\beta_T = 0.9$ and a tolerance time of $\tau_T = 20$ years. The $\pi_t^{tol}$ values for a system forced with the RCP4.5, RCP6.0 and RCP8.5 scenario's are

shown in Fig. 4 for both the monostable and bistable cases. As expected, the more extreme the RCP scenario, the earlier the PNR. This can be easily explained by the fact that when the $CO_2$eq concentration is rising faster, the temperature will get non viable earlier. Consequently, the PNR will be earlier, since the GMST is only allowed to be non viable for at most $\tau_T$ years. When the model is forced with RCP2.6, there is no PNR for both models. The reason for this is that the $CO_2$eq

concentration will remain low throughout the whole period and consequently the temperature will stay viable. The value of $\pi_t^{tol}$ of the bistable case is for each scenario earlier than the value of the monostable case. This can be clarified by the fact that the PDF of the temperature in the bistable case will leave the viable region at a lower $CO_2$eq concentration because of the existence of nearby equilibria.

The sensitivity of $\pi_t^{tol}$ versus the tolerance time $\tau_T$ and the tolerance probability $\beta_T$ was also investigated and the results are as expected (and therefore not shown). A longer tolerance time will shift $\pi_t^{tol}$ to later times. For example, for the RCP4.5 scenario $\pi_t^{tol} = 2071, \ 2088$ and $2116$ for





$\tau_T = 0, 20$ and $50$ years for the bistable case (for fixed $\beta_T = 0.9$). With a fixed $\tau_T = 20$ years, the value of $\pi_t^{tol}$ shifts to smaller values when the tolerance probability is increased. For example, for $\beta_T = 0.80$ and $0.99$, the values of $\pi_t^{tol}$ are $2127$ and $2058$, respectively, for the bistable case (for $\beta_T = 0.9$, it is $2088$, see Fig. 4).

## 4  PLASIM

The results in the previous section have illustrated that a PNR can be calculated when an estimate of the probability density function can be calculated and a collection of mitigation scenario's is available. We will now apply these concepts to the more detailed, high-dimensional, climate model PLASIM, a General Circulation Model developed by the University of Hamburg, and using mitigation scenario's derived from those suggested in the IPCC - AR5 report.

### 4.1  Linear response theory

In order to find the temporal evolution of the PDF of the global mean surface temperature GMST (indicated by $T$) under any $CO_2$eq forcing in PLASIM, we will use linear response theory (LRT). With this theory, the effect of any small forcing perturbation on the system state can be calculated by running the climate model for only one forcing scenario (Ragone et al., 2014).

In LRT, the expectation value of an observable $\Phi$, when forcing the system with a time-dependent function $f(t)$, can be calculated by computing the convolution of a Green's function $G_{\langle\Phi\rangle}$ and the forcing $f(t)$, according to

$$\langle\Phi\rangle_f(t) = \int\limits_{-\infty}^{+\infty} G_{\langle\Phi\rangle}(\tau) f(t-\tau) d\tau. \tag{9}$$

To construct this Green's function, the property that the convolution in the time domain is the same as point-wise multiplication in the frequency domain is used. The Fourier transform of Eq. (9) is given by

$$\langle\widetilde{\Phi}\rangle_f(\omega) = \chi_{\langle\Phi\rangle}(\omega)\widetilde{f}(\omega), \tag{10}$$

with $\chi_{\langle\Phi\rangle}(\omega)$, $\langle\widetilde{\Phi}\rangle_f(\omega)$ and $\widetilde{f}(\omega)$ being the Fourier transforms of $G_{\langle\Phi\rangle}(t)$, $\langle\Phi\rangle_f(t)$ and $f(t)$, respectively. Therefore, once the time evolution of the expectation value of an observable under a certain forcing is known, the Green's function of this observable can be constructed with Eq. (10) and consequently the linear response of the observable to any forcing can be calculated.

We use the same data as in Ragone et al. (2014), provided by F. Lunkeit and V. Lucarini (Univ. Hamburg, Germany). The only difference with those in Ragone et al. (2014) is that the seasonal cycle is not removed. This results in a long-term increase of the GMST of $5\ °C$ (instead of $8\ °C$ in Ragone et al. (2014)) under a scenario where the $CO_2$ concentration doubles. Data of GMST from two ensembles was used, each of 200 simulations, made with two different $CO_2$ forcing profiles




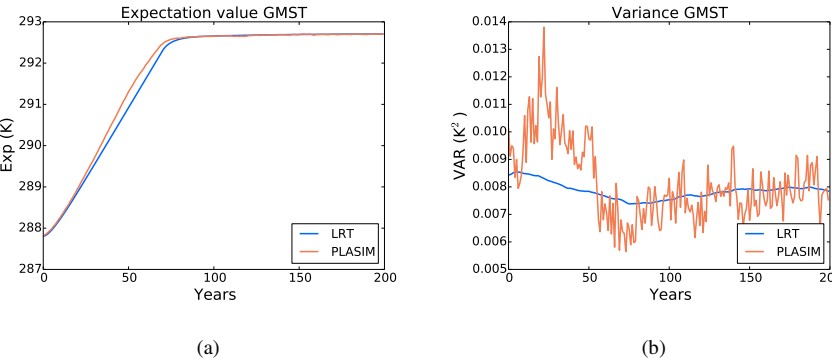

**Figure 5.** *(a) The expectation value and (b) variance of GMST generated by PLASIM (orange) and determined through LRT (blue).*

(all other GHGs are kept constant). For both forcing profiles, the starting $CO_2$ concentration is set to a value of 360 ppmv, which is representative for the $CO_2$ concentration in 2000. During the first set of experiments, the $CO_2$ concentration is instantaneously doubled to 720 ppmv and kept constant afterwards. During the second set of experiments, the $CO_2$ concentration increases each year with 1 % until a concentration of 720 ppmv is reached. This will take approximately

70 years and afterwards the concentration is fixed. The total length of the simulations is 200 years. Furthermore, the forcing $f(t)$ in Eq. (9) is the logarithm of the $CO_2$ concentration, since the radiative forcing scales approximately logarithmically with the $CO_2$ concentration.

    In order to determine the PDF of GMST under any $CO_2$eq forcing, we make the assumption that at each point in time the PDF of the GMST is normally distributed. As we have 200 data points

for the GMST at each time interval, a $\chi^2$ test was used to analyse the PDFs. For each time, the value of $\chi^2 > 0.05$ and therefore the assumption that the PDF of the GMST is normally distributed appears justified. The Green's functions for the expectation value and variance of GMST have been calculated with the instantaneously doubling $CO_2$ profile and the associated ensemble. From the ensemble, at each point in time the expectation value and variance are calculated such that we get

the temporal evolution of these two variables. Subsequently, we have found the Green's functions with Eq. (10). To check whether these Green's functions perform well, we compare the temporal evolution of the expectation value and variance of the GMST under the 1 % forcing (calculated with Eq. (9)) with those directly generated with PLASIM (Fig. 5). The expectation value made with LRT is close to the one directly generated by PLASIM. However, the variance of the ensemble generated

by PLASIM is a lot noisier than the one made with LRT. Although the Green's function of the variance provides only a rough approximation, it has the right order of magnitude and we will use it to calculate the variance of the GMST for other forcing scenarios.





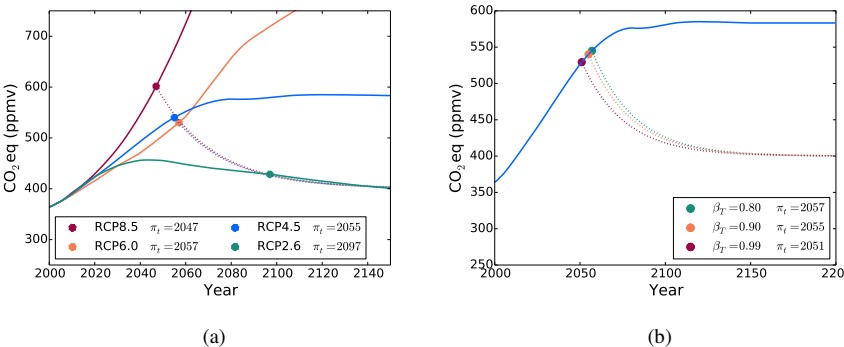

**Figure 6.** *(a) The PNR $\pi_t^{2100}$ for the RCP2.6, RCP4.5, RCP6.0 and RCP8.5 scenarios for a tolerance probability of $\beta_T = 0.9$. The solid lines represent the RCP scenarios and the dashed line the most extreme scenario from $F_\lambda$. Note that these dashed lines coincide. (b) The point of no return for RCP4.5 for different tolerance probabilities.*

### 4.2  Results: Point of No Return

The mitigation scenarios in $F_\lambda$ considered for use in PLASIM are exponentially decaying to different

stabilisation levels (varying between 400 and 550 ppmv Edenhofer et al. (2010)). This stabilization level is taken as the parameter $\lambda$. We assume that stabilisation happens within 100 years, which corresponds to an e-folding time $\tau_d$ of about 25 years; the mitigation scenarios $F_\lambda$ are then given by

$$F_\lambda(t) = \left(C_{t_c} - \lambda\right) \exp\left(-\frac{t - t_c}{\tau_d}\right) + \lambda, \tag{11}$$

where $t_c$ is again the time at which the mitigation scenario is applied and $C_{t_c}$ the associated $CO_2$eq

concentration. The most extreme mitigation scenario in $F_\lambda$ in terms of $CO_2$eq decrease is the one that stabilises at a $CO_2$eq concentration of 400 ppmv.

We next determine the PNR $\pi_t^{2100}$ by requiring that the GMST must be viable in 2100 using a tolerance probability of $\beta_T = 0.90$. Furthermore, the viable region is set at $T \leq 16.15°C$, which corresponds to temperatures lower than 2°C above the preindustrial GMST. The values of $\pi_t^{2100}$ for

all the RCP scenarios are plotted in Fig. 6a. The solid lines represent the RCP scenarios and the dashed lines present the most extreme scenario from $F_\lambda$. The value of $\pi_t^{2100}$ for the RCP8.5 forcing is 10 years smaller than that for the RCP6.0 scenario, since the $CO_2$eq concentration increases much faster for the RCP8.5 scenario. However, the mitigation scenario after the point of no return, represented by the dashed line, is for all RCP scenarios the same. This is related to the definition of

$\pi_t^{2100}$, where it is required that the GMST is viable in 2100. The mitigation scenario that is plotted is the ultimate scenario that guarantees this. It indicates that (if the mitigation scenario plotted in Fig. 6a is extended for years smaller than 2047) for each $CO_2$ scenario the associated $\pi_t^{2100}$ is given by the intersection of that $CO_2$eq scenario and the mitigation scenario. This is because it is considered





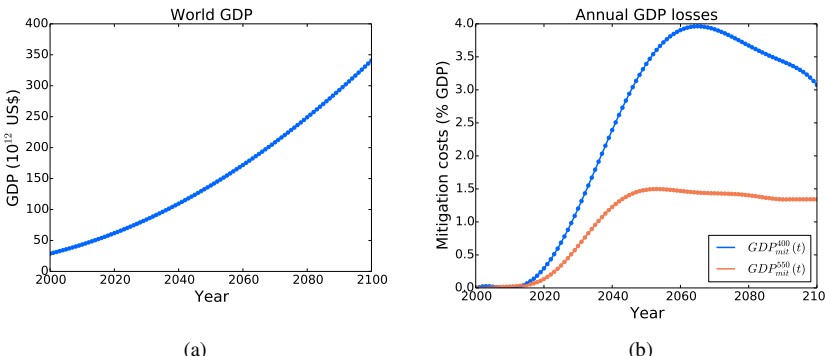

(a)                         (b)

**Figure 7.** *(a) The expected rise of GDP expressed in US dollars and (b) the annual economic costs of stabilising after 100 years at $CO_2eq$ levels of $400$ ppmv and $550$ ppmv. Both figures are from Edenhofer et al. (2010) and show results of the POLES model. Note that to calculate the annual costs of stabilising at another $CO_2eq$ level, for each year we linearly interpolate the costs.*

that an exponential decay to 400 ppmv within 100 years is always possible, no matter the $CO_2eq$

concentration at $t_c$. However, when this concentration becomes too high, this mitigation scenario is not realistic anymore (as discussed in the next subsection).

The influence of the tolerance probability on $\pi_t^{2100}$ for the RCP4.5 scenario is plotted in Fig. 6b, where we only consider a tolerance probability of 0.8, 0.9 and 0.99. When the tolerance probability is higher, it takes longer before the GMST will be viable again and thus the PNR will be earlier.

However, the differences are very small, since the mitigation scenarios that guarantee viability in 2100 for the different tolerance probabilities are very close to each other.

**4.3 Results: Optimal mitigation scenario**

The approach followed here also provides the possibility to determine an optimal mitigation strategy as each of these mitigation scenario's has its own economic costs. The costs associated with the

stringency of mitigation are determined using the output of the POLES model. This is an energy-environment-economy model and used by Edenhofer et al. (2010) to calculate the economics of low $CO_2eq$ stabilisation. Fig. 7(a) reveals how the POLES model expects the global gross domestic product (GDP, below indicated by $\mathcal{G}$) to increase till 2100. The annual costs of stabilising at $CO_2eq$ levels of $400$ ppmv and $550$ ppmv are shown in Fig. 7(b) (Edenhofer et al., 2010); the costs are

expressed in percentage of the GDP. The cost for stabilising at $400$ ppmv are a lot higher than for stabilising at $500$ ppmv. This is because such a low stabilisation can only be reached when new technologies, like energy made out of biomass and carbon storage, are used on a large scale.

In order to calculate the economic costs of a mitigation scenario that stabilises at a certain $CO_2eq$ level, the following assumptions are made.





(i) The annual costs of a mitigation scenario that stabilises at another $CO_2$eq level than $400$ or $550$ ppmv are linearly interpolated from the costs at these two values.

(ii) Costs are only dependent on the stabilization level and are independent of the $CO_2$eq concentration at $t_c$ ($C(t_c)$). To determine the dependence on $C(t_c)$ would require an analysis of different mitigation scenarios than in Edenhofer et al. (2010) and this is beyond the scope of this study.

(iii) Costs are taken into account only for mitigation between $t = t_c$ and $t = 2100$.

The annual GDP loss associated with a scenario that stabilises at $\lambda = C_{st}$ is indicated by the function $\mathcal{G}_{mit}^{\lambda}(t)$.

When the temperature reaches a value of $2°$C above the preindustrial temperature, Stern (2007) estimated the annual costs associated with damage of extreme weather events at 0.5-1.0% of the GDP. We have chosen here to charge 1.0% GDP ($\mathcal{G}_{ewe}$ below) for every year the GMST is not viable; when the temperature is viable, no costs will be charged. Consequently, the costs associated with the stringency of mitigation for a mitigation scenario are given by

$$\Psi\big(F_\lambda(t)\big) = \frac{1}{100}\Big(\sum_{t=t_c}^{2100} \mathcal{G}(t)\, \mathcal{G}_{mit}^{\lambda}(t) + \sum_{t=t_b}^{t_e} \mathcal{G}(t)\, \mathcal{G}_{ewe}\Big) \tag{12}$$

The optimal mitigation scenario can be found by minimising the cost function $\Psi$ under the restriction that the climate must be viable in 2100. As an example, we consider the forcing according to the RCP4.5 scenario, where the first year of non viability is 2027. For $\Delta t = 4$ years, the year when mitigating is started is 2031. In Fig. 8a, the costs are plotted for the different mitigation scenarios. Stabilisation concentrations of 425 ppmv and higher are not considered, since these scenarios do not guarantee that the GMST is viable in 2100. The optimal mitigation scenario is a scenario that stabilises at a $CO_2$eq concentration of 412.4 ppmv. The wiggles in Fig. 8a can be explained by the fact that for some scenarios the amount of years of non viability are the same. However, the period of the wiggles gets smaller as the stabilisation $CO_2$eq concentration increases. To explain this, we note that the moment of getting viable again ($t_e$) corresponds with a certain reference $CO_2$eq concentration. During low stabilisation, the decrease in $CO_2$eq concentration is steep and there is not much time between the intersection of two consecutive mitigation scenarios with this reference concentration. However, when the stabilisation concentration is higher, the decrease in $CO_2$eq concentration will be less steep, and there is more time between the intersection of two consecutive scenarios and the reference concentration.

Fig. 8b shows how the costs of the optimal mitigation scenario change when $\Delta t$ increases. The results in Fig. 8a are for $\Delta t = 4$ years and $\Delta t = 28$ years corresponds with the point of no return. Fig. 8b reveals that there is a $\Delta t$ that maximises the costs of the optimal mitigation scenario. When $\Delta t$ increases, the time between $t_b$ and $t_e$ will increase and thus the costs associated with extreme weather events increases. Also, when $\Delta t$ increases, the optimal mitigation scenario is steeper and





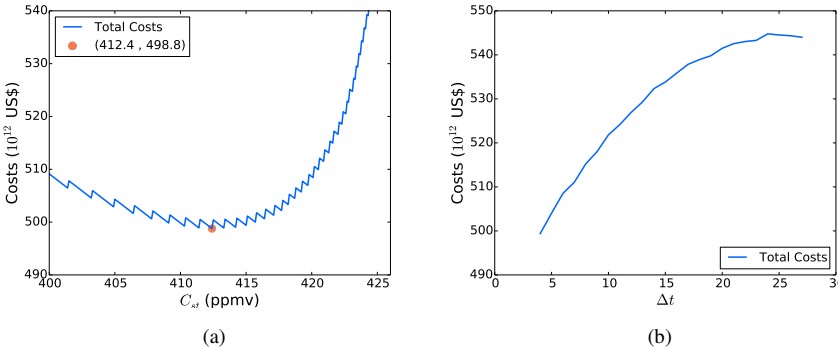

(a)                                        (b)

**Figure 8.** *(a) The costs for the different mitigation scenarios for the case of RCP4.5, $\beta_T = 0.9$ and $\Delta t = 4$ years. On the horizontal axis is the stabilisation $CO_2eq$ level of the scenarios in $F_\lambda$. The optimal mitigation scenario is shown by the orange dot. The associated costs can be found in the legend. (b) The costs of the optimal mitigation scenario for different $\Delta t$. The point of no return $\pi_t^{2100}$ corresponds to $\Delta t = 28$ years.*

thus more expensive. However, since the time between $t_c$ and 2100 will decrease as well, the total costs associated with the stringency of mitigation eventually decrease for larger values of $\Delta t$.

## 5  Discussion

Pachauri et al. (2014) stated with high confidence that: "Without additional mitigation efforts beyond those in place today, and even with adaptation, warming by the end of the 21st century will lead to
high to very high risk of severe, widespread and irreversible impacts globally". If no measures are taken to reduce GHG emissions during this century and neither will there be any new technological developments that can reduce GHGs in the atmosphere, it is likely that the GMST will be 4 °C higher than the preindustrial GMST at the end of the 21st century (Pachauri et al., 2014). Consequently, it is important that anthropogenic emissions are regulated and significantly reduced before widespread
and irreversible impacts occur. It would help motivate mitigation when we would know when it is "too late".

   In this study we have defined the Point of No Return (PNR) in climate change more precisely, using stochastic viability theory and a collection of mitigation scenario's. For an energy balance model, as in section 3, the probability density function could be explicitly computed and hence
stochastic viability kernels were determined. In this model, the PNR was based on a tolerance time for which the climate state is non viable. For the RCP scenario's considered, one finds that the PNR is smaller in the bistable than in the monostable regime of this model. The occurrence of possible transitions to warm states in this model indeed cause it to be 'too late' earlier.

   Key innovation in our approach, however, is to use linear response theory (LRT) for high-dimensional
models to estimate the probability density function. We showed this by computing PNRs for the



PLASIM model (section 4), where the PNR is based on only requiring that the climate state is viable in the year 2100. In the PLASIM results, we used a viability region that was defined as GMSTs lower than 2°C above the pre-industrial value. This 2°C level gained a lot of prominence in the early 1990s when a number of international scientific panels suggested that it would prevent some of the

worst impacts of climate change. Recently, during the 2015 Paris Climate Conference (COP21), it was decided that 1.5°C is a significantly safer threshold. With our methodology, the PNR can be easily determined for any threshold defining the viable region.

Although our approach provides new insights into the point of no return in climate change, we recognize there is potential to substantial further improvement. The assumption about the most ex-

treme scenario from $F_\lambda$, here considered to be an exponential decay to 400 ppmv within 100 years, is too simplified. Once, the $CO_2$eq concentration at $t_c$ is very high, this is an unrealistic scenario. Therefore, further research must be done to determine the most extreme mitigation scenario for each $CO_2$eq concentration. We also designed a rather idealized cost function to find the optimal mitigation scenario in PLASIM. The costs associated with the stringency of the mitigation scenario are calcu-

lated using the POLES model. In this model the economic costs are given for one specific $CO_2$eq scenario that stabilises at 400 ppmv. However, we use these costs to calculate the costs for any scenario that stabilises at 400 ppmv, no matter the $CO_2$eq concentration at $t_c$. This is a shortcoming in our cost function which can be improved by using the POLES model to calculate the costs for each separate mitigation scenario.

Another shortcoming in the cost function lies in the part that calculates the costs associated with extreme weather events. We only charge costs for extreme weather events when the GMST is not viable. Also, for each year that the GMST is not viable, no matter the height of the GMST, the costs are given by the same percentage of the GDP. Furthermore, we only consider costs associated with stringency of mitigation and extreme weather events. However, there are a lot more factors

that influence the economic costs, for example, sea level rise, agriculture and human health. A more realistic cost function could probably be designed based on Stern (2007) but is beyond the scope of this study.

Due to these shortcomings, one cannot attribute much importance to the precise PNR values obtained for the PLASIM model. However, we think that our approach is general enough for han-

dling many different political and socio-economical scenario's combined with state-of-the-art climate models. Hence, it will be possible to make better estimates of the PNR for the real climate system. We therefore hope that eventually these ideas on the point of no return in climate change will become part of the decision-making process during future debates about climate change.

*Acknowledgements.* This study was supported by the MC-ITN CRITICS project (no. 64307). We thank Valerio

Lucarini and Frank Lunkeit (Univ. Hamburg) to provide the PLASIM model data.





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
