# Peer review of "On determining the Point of no Return in Climate Change"

_Earth System Dynamics, 2016_

## Referee Comment (RC2)

**Referee comment from Kristoffer Rypdal**

**K. Rypdal**

Department of Mathematics and Statistics, UiT The Arctic University of Norway, Norway

Correspondence to: Kristoffer Rypdal (kristoffer.rypdal@uit.no)

70

**1 General comments**

I agree with most comments of Referee #1, although I don't think the issue of tipping points is so serious. I think it is acceptable to do the analysis under the assumption that no

- 5 major tipping points will be attained in the relevant period, since any analysis of this type would be impossible in that case, unless one has a completely reliable nonlinear model of the climate system. And, the authors *do* in fact make an attempt to include the effect of nonlinearity and the pres-
- ence of a tipping point in their analysis of the energy balance model (EBM). In their bistable version of the EBM, they find some effect on the point of no return (PNR) from the reduced stability due to the presence of nearby equilibria. My main criticism here is not that they don't recognise the impact of 50 tipping points, but the lack of realism of their EBM.
- The first paragraph of the report of Referee #1 gives a brief description and assessment of the goals and methods of the paper, which I find no reason to repeat here. Referee #1 points out the presence of non-recognised key simplifying
- assumptions as a major weakness of the paper. There are, however, other and more serious key simplifying assumptions that are not recognised and discussed in this paper, and which strongly limits its potential utility. And worse, some may be right out misleading and harmful if they are adopted 60
  by policy makers.
  - The most serious assumption is the totally unrealistic form of the mitigation scenarios used, which ignore the existence of a carbon cycle. The authors assume implicitly that the atmospheric  $CO_2$  concentration can be manipulated directly.
- 30 One consequence is that the point of no return (PNR) by definition in this paper always occurs *after* the climate state has become non-viable. In the real climate system, the PNR will occur *before* the climate has become non-viable, which makes a profound difference.
- The EBM employed actually describes the Earth's climate in an icehouse state. It should be replaced by a more realistic model that displays bistability, e.g., a model for the small icecap instability.

The analysis of the cost function is inadequate, and a correct analysis would reveal the shortcomings of assuming that the mitigation action comes after the viability limit is breached.

**2 Specific comments**

**2.1 How to use simplified models**

There is nothing wrong in using super-simple models for the global climate response in studies of the type presented in this paper. In fact, apart from an enormous reduction of computational cost, well chosen conceptual models are often more correct than some general circulation models (GCMs) in projecting the global mean surface temperature (GMST). In Rypdal and Rypdal  $(2014)^1$  we show that a simple, linear response model (with a power-law Green's function and deterministic and stochastic forcing components) give results for the GMST in the instrumental period that are indistinguishable from those of the CMIP5 archive (Fig. 15 in that paper). The realism of the model is established by the powerlaw form of the Green's function which reflects the longrange memory (LRM) in the climate response and that the parameters of the model are estimated from observational data for radiative forcing and instrumental GMST. The EBM employed in this paper as well as the PLASIM GCM lack this memory in the response. Both operate with a mixedlayer ocean which yield an exponentially decaying impulse response of the GMST with time constant from a few years to a decade or two. This makes them miss the "warming in the pipeline" associated with the heat transport from the mixed layer into the deep ocean. By leaving out this delay in the GMST response to forcing, the GMST will decay faster after atmospheric  $CO_2$  concentration is reduced. The effect is illustrated in Fig. 1 below, where the GMST in a full-blown

<sup>1M. Rypdal and K. Rypdal, Long-memory effects in linearresponse models of Earth's temperature and implications for future global warming, J. Climate, 27, 5240-5258, doi:10.1175/JCLI-D-13-00296.1, 2014

atmospheric-ocean GCM fails to stabilise at a new equilibrium even centuries after a step-function rise in  $CO_2$  concentration. In contrast, the PLASIM model would stabilise at a new constant GMST after a few years.

**Figure 1.** Grey curve is the global temperature response to a sudden 4-doubling of atmospheric CO2 concentration in the GISS-E2-R model. Blue curve is a fit of superposition of two exponential responses (two-box model solutions); the two exponential time constants being  $\tau_1 = 1.3$  yr and  $\tau_2 = 176$  yr. Red curve is a power-law 100 fit, and is a poor fit up to several years, but a good fit in longer time scales. Note that temperature will continue to rise for hundreds of years.

- An even more serious flaw, however, is the implementation of mitigation scenarios in the form of exponentially decaying  $CO_2$  concentration with e-folding time of 9 and 25 years. This is at odds with anything we know about the carbon cycle and with any realistic emission scenario, even with car-
- 80 bon capture and storage (CCS) implemented on a scale that is economically feasible. I recommend the authors to take a look at my recent ESD paper2 where I study CO2 and GMST projections for some idealised (but realistic) emission scenarios by very simple models for the responses. The emission
- scenarios shown in Fig. 2, display the annual emissions when mitigation measures come into action in years 2030, 2070, and 2110, respectively. The base scenario (blue curve) is a business as usual (BAU) scenario that continues the presentday exponential rise. It is close to the RCP8.5 scenario up
- to 2070. The mitigation actions considered are an annual reduction of emissions of 1% and 5%, respectively. Economic studies indicate that higher annual reductions than 5% can not be attained without disrupting the global economy, while 1% seems to be a realistic upper limit. 5% reduction corresponds to an e-folding time of about 14 years, while 1%
- corresponds to 70 years.

The crucial point, however, is that an exponential decay 1 of the emission rate does not correspond to an exponential decay of the atmospheric  $CO_2$  concentration due to the long

**Figure 2.** Blue curve is carbon emission rate R(t) scenario obtained by fitting the exponential  $S_0 \exp gt$  to the emission rate 4 GtC/yr in 1960 and 11 GtC/yr in 2010 AD. The full, brown, orange, and red curves are the subsequent R(t) after initiation of 1% reduction of emission rate per year. The dashed curves are corresponding rates with 5% reduction per year.

residence time of atmospheric  $CO_2$ . The modeled evolution of the  $CO_2$  concentration from the described emission scenarios is shown in Fig. 3. It is apparent that these mitigation scenarios (even in the extreme 5% reduction case) are far less radical than those considered by the authors of the paper under review. In fact, for economically and politically realistic mitigation scenarios (1% annual reductions of emissions), the  $CO_2$  concentration will continue to rise monotonically beyond year 2200.

105

**Figure 3.** Projections of  $CO_2$  concentration under the emission scenarios in Fig. 2 using the simple response model for  $CO_2$  proposed in K. Rypdal (2016)2.

When memory effects in the GMST response are taken into account the GMST projections look even more bleak. If 2° GMST rise is taken as the viability limit, none of the emission scenarios I have considered will prevent a monotonically rising GMST beyond year 2200 if mitigation action is taken at a time  $t_c$  after the time  $t_b$  when this limit is breached. Hence, by assuming that  $t_c > t_b$  there is no useful way to define the PNR. It will not exist for a realistic set

<sup>2K. Rypdal, Global warming projections derived from an observation-based minimal model, Earth Syst. Dynam.,7, 51-70, 115 2016, doi:10.5194/esd-7-51-2016

155

160

170

**Figure 4.** Projections of GMST under the  $CO_2$  concentration scenarios shown in Fig. 2, using a linear model for the GMST response proposed in K. Rypdal (2016)2.

of mitigation scenarios unless one defines the tolerance time  $\tau_T$  to be several centuries, which is, of course, of no interest to policy makers. One may argue of course, that the projections I present in Figs. 3 and 4 are too pessimistic. However, in the ESD-paper2 I present also some projections that most probably are too optimistic (atmospheric CO2 half-life of 33 years, and weak memory in the GMST response). But also with these projections the GMST will continue to rise beyond year 2100 if  $t_c > t_b$ .

My conclusion is that if the authors want to deal with the real world they should employ more realistic mitigation scenarios. Mitigation measures should indicate emission, not atmospheric concentration, and some model for the carbon

- 130 cycle is necessary. Models for the GMST response to CO2 forcing should take delayed responses into account. A two-box model including the deep ocean in addition to the mixed layer is an alternative to the power-law response used by us.1 The PLASIM model is not adequate for this purpose, and the
- 135 CMIP5 model ensemble is too small to be useful for a statistical study.

**2.2 Science-fiction scenarios**

120

125

One can always discuss the utility of studying scenarios that seem impossible with today's technologies, such as rapid de-

- 140 pletion of the atmospheric  $CO_2$  content, and super-optimistic assumptions about the memory in the GMST response. But a minimum requirement in a paper like this is a discussion of 175 the realism of the mitigation scenarios considered. As an input to such a discussion I show in Figs. 5 and 6 the radiative
- 145 forcing and the GMST resulting from abrupt, discontinuous transitions to a zero emission regime in years 2030, 2070, and 2110, respectively (in these plots zero time corresponds 180 to year 1880). Note that in Fig. 5 I plot the forcing, and not the CO2 concentration. The model used to produce these pro-
- 150 jections assumes that, after emissions are cut to zero, a certain fraction of excess atmospheric CO2 content is removed

from the atmosphere, such that the concentration decays exponentially towards preindustrial level with e-folding time of 33 years. This time constant was estimated from the historic emission record and the Mauna Lua CO2 concentration record, using the exponential-decay model.2 It is well known from carbon-cycle models that the exponential decay is unrealistic on time scale of centuries and longer because the uptake in oceans and vegetation will saturate, but this model can serve as a sci-fi limit. The model also assumes a memory exponent  $\beta = 0.35$  for the GMST response which is half the value estimated from instrumental data.1

We observe form Figs. 5 and 6 that even in this totally unrealistic scenario the e-folding time for the forcing is around 50 years, and for GMST about 70 years. In comparison, the  $CO_2$  concentration and GMST scenarios studied by the authors in Section 4.2 have e-folding time of 25 years.

**Figure 5.** Projections of  $CO_2$  forcing under emission scenarios with abrupt (step function) transition to zero emission, using an exponential-decay model with e-folding time 33 yr for  $CO_2$  concentration.

**2.3 The energy balance model**

The Budyko EBM employed by the authors, with a temperature-dependent albedo as the main nonlinearity, is useful to illustrate the possibility of an abrupt transition between a "icehouse Earth" state and a "greenhouse Earth" state with small or no icecaps. The authors consider two albedo profiles one with  $\alpha_1 = 0.2$  in the warm state, and one with  $\alpha_1 = 0.45$  in the warm state. In both cases  $\alpha_0 = 0.7$  in the cold state. For the former the model yields on one stable branch with GMST of about 255 K. This monostable model is unrealistic because of the high albedo  $\alpha_1$  chosen for high GMST. There is also no explanation for why the emissivity is set to  $\epsilon = 1$ , while the true value is around 0.6. If  $\alpha_1$  is reduced beyond a certain point there will also be a warmer stable branch and the two stable branches are connected by an unstable branch. The authors have decided that they want the present-day climate to reside on the cold branch, so they sim-

---

## Referee Comment (RC1) · Anonymous Referee #1 · 25 Sep 2016

This manuscript describes a method to determine the time at which greenhouse gas emissions must be reduced in order to avoid crossing a temperature threshold with a given probability. The concepts of the method are articulated and it is tested in an energy balance model. The method is then employed in a GCM with some further simplifying assumptions. Finally, a so called 'optimum mitigation scenario' is described based on further simplified cost assumptions.

The work is mostly novel and is partly well described and explained. However, some of the key simplifying assumptions are not recognized or explained, and these seriously undermine the potential utility of the method. These limitations should be acknowledged, along with their implications for use of the method to define a point of no return. Some of these limitations are described below.

**1. Climate system not well behaved**

The major limitation of the method is that it assumes that the climate is mostly well behaved without tipping points. The method assumes that once the state vector Xt is 'not viable', then greenhouse control measures can be introduced to render Xt viable again at a future point in time. That assumes that everything is reversible and that there are no tipping points crossed or critical impacts incurred during the period when Xt is not viable. For example, tipping points may exist in the oceans or cryosphere where circulation or melt processes persist even when the control measure is taken. Alternatively, critical impacts may occur which one would wish to prevent, but which won't be prevented by the control measure.

The effect of these nonlinearities and impacts is that the actual point of return may be much earlier than the one calculated here where the climate is well behaved and just has to be returned below the threshold global mean temperature change value. The failure to recognize and acknowledge this limitation is critical for the method, since it means that the method is somewhat reckless in ignoring the potential for abrupt, irreversible and/or persistent changes, and would furnish policy makers with a false sense of security about the alleged point of no return. The point of return calculated for this method is for a planet with no irreversible or very persistent changes, which is almost certainly violated for the actual planet. The authors propose their method for use in policy, which is an over-reach given the critical, unacknowledged limitations of the method.

**2. GMST normally distributed**

The extension of the method here to GCMs assumes that GMST is normally distributed. That may be the case for some GCMs and it may be the case when the climate is not undergoing a transition. However, departures from Gaussianity will occur when the climate state is undergoing transition, which is an example of the kind of nonlinear response noted in point 1 above that is not captured by the method here. The introduc-

tion of the normal distribution assumption here further predicates the method on a well behaved climate system. By using GCMs and assuming that the response is always Gaussian the method underestimates the potential for changes in the climate system that violate the assumptions of the method.

3. Optimal mitigation scenario

The approach followed to determine the optimal mitigation scenario takes a very narrow view of only some costs and only economic costs. This results in potentially misleading estimates of what is 'optimal' and what isn't. To do justice to this issue would require much more serious analysis and exploration of uncertainty than is carried out in this paper. As it stands, this section of the paper is so flawed and dependent on the precise set of assumptions used that it would need to be removed from the paper.

---

## Author Comment (AC1) · 24 Nov 2016

We thank the reviewer for the useful comments on the manuscript and a point-by-point reply follows below. The aim of the manuscript is to introduce the new methodology combining stochastic viability theory, linear response theory and economic modeling to address the concept of the Point of no Return $\pi_t$. We indeed realize that the models used are only illustrative of the methodology and not aimed to provide a realistic estimate of $\pi_t$. Although this was mentioned in the original manuscript (in the discussion), it will be mentioned much more explicitly in the revised manuscript.

**1: Climate system not well behaved**
**Reply:**

In section 3 of the paper we show (using a simplified energy balance model (EBM)) that in principle tipping points can be taken into account to determine $\pi_t$ once the PDF of the GMST can be computed (as can be done for the EBM). However, when linear response theory is used to compute PDFs, such as in section 4 for the PLASIM, indeed tipping behavior cannot be captured. This will now be mentioned more explicitly in the revised discussion of the paper.

**2. GMST normally distributed**
**Reply:**

This is indeed an assumption limiting the applicability of the methodology. The PDFs of PLASIM are approximately Gaussian but this is not expected to hold for any climate model and in particular when transitions do occur, PDFs are not Gaussian. Again this will be mentioned in the revised discussion.

**3. Optimal mitigation scenario**
**Reply:**

We do agree that this view on the economic costs of mitigation is simplified although we directly use results from the POLES model regarding the stabilization scenarios (Edenhofer et al. 2010). However, by the recommendation of both reviewers, we will delete section 4.3 of the paper, but we will mention the possibility of determining optimal mitigation scenarios in the revised discussion.

---

## Author Comment (AC2) · 24 Nov 2016

We thank the reviewer for the detailed report and the very useful comments on the manuscript. A point-by-point reply follows below, where we refer to each section of the report.

**4. Recommendation**
**Reply:**

The aim of the paper is to introduce a novel methodology, combining stochastic viability theory, linear response theory and economic modeling to address the concept of the point of no return $\pi_t$. Indeed, the models employed (even the PLASIM), the mitigation

scenarios and the cost function are highly idealized. However, they are used here to illustrate the methodology and not to provide a realistic estimate of $\pi_t$. Although this was mentioned in the original manuscript (in the discussion), we agree that we should provide a much more critical discussion and we will do so in the revised paper. Determining the point of no return in a realistic setting is the next step on which we are currently working and is basically a new paper.

**2.4 The model for and analysis of the cost function**
**Reply:**

Also based on the recommendation of reviewer #1, the original section 4.3 will be deleted in the revised paper. Instead we will mention the possibility of determining optimal mitigation scenarios in the revised discussion.

**2.3 The energy balance model**
**Reply:**

This model is indeed constructed to illustrate the methodology in the presence of a saddle-node bifurcation. We think that it serves this purpose, without claiming to be realistic here (which is questionable anyway with these type of models). In the revised version, this will be explicitly mentioned.

**2.2 Science-fiction scenarios**
**Reply:**

We agree but this is an additional detail which is not needed to illustrate the concept

of the point of no return and the methodology to determine it. However, in the revised section 4.2, we will discuss the effect of the value of the e-folding time scale (for the PLASIM) on the results.

**2.1 How to use simplified models**
**Reply:**

Our opinion is that models should always be targeted to the question which is asked. Idealized models can be used when answering questions on mechanisms and illustrating methodology, but it is questionable whether such simplified models can determine relations between relevant observables (here radiative forcing and the global mean temperature). Anyway, our motivation here is to use these models only to illustrate the methodology and concepts and in that respect this is justified.

**1. General comments**
**Reply:**

The only remaining issue is the fact that the point of no return can indeed occur before the climate becomes non-viable due to the delay of the carbon cycle response. We will address this issue in the revised discussion.

**3. Some minor points**
**Reply:**

**Page 6, lines 163–167:** Suggestion will be followed and the value of $\sigma^2$ will be mentioned.

[Figure]

**Figure 3:** More information will be provided to explain the results of this figure.

**Page 10, lines 261–263:** We will add a paragraph to explain this difference.

**Page 15, line 384:** We were not aware of the Rypdal (2015) paper where this suggestion is made. We will compare this approach with the one based on linear response theory and add the results in the revised paper.

---

## Referee Report (RR1)

The study introduces a number of important concepts and a potentially very powerful framework for policyrelevant analysis. I also enjoyed the hierarchical approach with the use of a Budyko-Seller model to illustrate the methodology.

On the other hand, there are some aspects of the analysis and discussion that I believe need to be altered or addressed in greater detail. I would recommend the paper for publication in ESD provided the authors implement the appropriate changes.

**Major Comments:**

- My main concern is the applicability of the methodology to CMIP5 (and soon CMIP6) simulations. It
  is unrealistic to expect 200-member ensembles of CMIP5 models to become the norm anytime soon.
  What conclusions could the authors actually draw from the simulations currently stored in the CMIP5
  archive or those already being run/planned for CMIP6? I am looking for a detailed, critical evaluation
  of the real applicability of the methodology. Just adding a one-line statement will not be sufficient.
- While reading the paper, I often found the wording a bit awkward (a non-exhaustive list: II. 53 and following, II. 67 (Giving → Given?), I. 143 (non-viable for?), II. 176 and following etc. ). I would encourage the authors to carefully revise the text as part of their revisions.
- 3. In order to ensure an unbiased approach to the study, I did not read the comments of the other Reviewers until after having noted down my own. However, I generally agree with the feedback provided in the first two reviews, and note that the authors could have addressed more extensively some of the points that were raised. In fact, in many places the authors seem to have performed a minimal-effort revision. For example, Reviewer #1 correctly questions the assumption of Gaussianity of the GMST. In my opinion, a brief mention of this limitation in the discussion section is insufficient. What do instrumental/reanalysis datasets tell us about the GMST distribution in the recent/present climate? Since, if the authors themselves or others will (hopefully) apply this method in the future, GMST might not be the only variable of interest. How easily can the method be adapted to other types of distributions? Similarly, Reviewer #2 highlights a potentially important improvement that could be applied to the methodology (Page 15, I. 384), but the authors seem to have taken little notice of this. This could also help to address my comment #1 above, which I think is one of the crucial limitations of the method.

**Minor points:**

1) There are lots of acronyms in the abstract, which make it unnecessarily unwieldy. While it is perfectly fine to use acronyms in the paper, I would encourage the authors to limit their use in the abstract.

2) I. 41 Explain what the Burning Embers diagram is or remove the reference altogether.

**3) I. 46 Define pCO2.**

4) The methodology is spread across the different sections of the paper. While in some cases this works well in terms of logical flow, it makes it very difficult for the reader to obtain an overview of what type of analysis is being carried out. For example, the authors could consider moving some of the more technical parts of Section 4.1 to the Methodology Section.

5) The description of the PLASIM model is very dry. A few more details would help the reader to better understand the type of data being used here without having to refer to other papers.

---

## Author Response (AR2)

**MS-No.:** esd-2016-40

**Version:** Second revision

**Title:** On determining the point of no return in climate change

**Author(s):** van Zalinge, Feng, Aegenheyster and Dijkstra

**Point by point reply to reviewer #3**

**June 21, 2017**

We thank the reviewer for the very useful comments on the manuscript. A point-by-point reply to all comments follows below.

**Major Comments**

*1. My main concern is the applicability of the methodology to CMIP5 (and soon CMIP6) simulations. It is unrealistic to expect 200-member ensembles of CMIP5 models to become the norm anytime soon. What conclusions could the authors actually draw from the simulations currently stored in the CMIP5 archive or those already being run/planned for CMIP6? I am looking for a detailed, critical evaluation of the real applicability of the methodology. Just adding a one-line statement will not be sufficient.*

**Reply:**

We have recently applied the same methodology to two CMIP5 ensembles of models, i.e. a 34 member control and abrupt 4x $CO_2$ forcing ensemble and a 35 member control and 1% $CO_2$ increase ensemble. The 4x $CO_2$ forcing ensemble was used to derive the Green's function and then the 1% $CO_2$ increase ensemble was used as a check on the resulting response. Although the ensemble is relatively small and the models within the ensemble are different (but many are related), the results (shown in the Fig. 1) are surprisingly good. The probability density function of GMST increase is close to Gaussian for the 1% $CO_2$ increase ensemble but clearly deviates from a Gaussian distribution for the 4x $CO_2$ forcing ensemble, particularly at later times. However, this indicates that the methodology can be applied successfully to results of simulations of CMIP5 (and in the future CMIP6) models. We mention these new CMIP5 results now briefly in the revised discussion.

*2. While reading the paper, I often found the wording a bit awkward (a non-exhaustive list: ll. 53 and following, ll. 67 (Giving → Given?), l. 143 (non-viable for?), ll. 176 and following etc. ). I would encourage the authors to carefully revise the text as part of their revisions.*

**Reply:**

We have again gone carefully through the manuscript and have corrected these and additional style errors.

[Figure]

Figure 1: *Ensemble mean (left panel) and variance (right panel) of GMST response (solid curves) and those determined through linear response theory (dashed curves). Blue curves are the ones from the ensemble of 4x $CO_2$, and the green ones are from the ensemble of 1% $CO_2$ increase.*

*3. In order to ensure an unbiased approach to the study, I did not read the comments of the other Reviewers until after having noted down my own. However, I generally agree with the feedback provided in the first two reviews, and note that the authors could have addressed more extensively some of the points that were raised. In fact, in many places the authors seem to have performed a minimal-effort revision. For example, Reviewer #1 correctly questions the assumption of Gaussianity of the GMST. In my opinion, a brief mention of this limitation in the discussion section is insufficient. What do instrumental/reanalysis datasets tell us about the GMST distribution in the recent/present climate? Since, if the authors themselves or others will (hopefully) apply this method in the future, GMST might not be the only variable of interest. How easily can the method be adapted to other types of distributions? Similarly, Reviewer #2 highlights a potentially important improvement that could be applied to the methodology (Page 15, l. 384), but the authors seem to have taken little notice of this. This could also help to address my comment #1 above, which I think is one of the crucial limitations of the method.*

**Reply:**

In the revised section 5, we now discuss the assumption of Gaussian GMST distribution, also with reference to the CMIP5 results.

The applicability of the methodology to other observables than GMST can in principle be performed but can be less useful than for GMST (as is now also mentioned in section 5).

We tried to apply the power-law response function for GMST (suggested by referee #2 of the previous version), but that did not give any useful results (as is now mentioned in the revised section 4.1).

**Minor points**

*1. There are lots of acronyms in the abstract, which make it unnecessarily unwieldy. While it is perfectly fine to use acronyms in the paper, I would encourage the authors to limit their use in the abstract.*

**Reply:**

Thes acronyms have been removed from the abstract.

*2. l. 41 Explain what the Burning Embers diagram is or remove the reference altogether.*

**Reply:**

We do not mention the Burning Embers diagram anymore.

*3. l. 46 Define pCO2*

**Reply:**

We removed pCO2 from the paper, as it was confusing. We now use $CO_2$ concentration.

*4. The methodology is spread across the different sections of the paper. While in some cases this works well in terms of logical flow, it makes it very difficult for the reader to obtain an overview of what type of analysis is being carried out. For example, the authors could consider moving some of the more technical parts of Section 4.1 to the Methodology Section.*

**Reply:**

We moved the paragraph on linear response theory (LRT) to section 2 (new section 2.2).

*5. The description of the PLASIM model is very dry. A few more details would help the reader to better understand the type of data being used here without having to refer to other papers.*

**Reply:**

We added a paragraph on the description of the PLASIM model at the beginning of the revised section 4.

**MS-No.:** esd-2016-40

**Version:** Second revision

**Title:** On determining the point of no return in climate change

**Author(s):** van Zalinge, Feng, Aegenheyster and Dijkstra

**Point by point reply to reviewer #4**

**June 21, 2017**

We thank the reviewer for the very useful comments on the manuscript. A point-by-point reply to all comments follows below.

**Issues Raised**

*1. The authors claim to apply some form of viability theory for their analysis, but their presentation and usage of viability theory is confusing if not misleading. Viability theory is mainly about control problems in which the trajectory of some system can be influenced at many time points rather than just once as in this paper. In viability theory, the (stochastic) viability kernel of a system given control options and a desirable region in state space consists of all states starting from which one can force the system to stay within the desirable region (with at least some given probability) forever or until some target time by using a suitable (and typically non-constant!) control policy. The (somewhat unclear) definition of viability kernel the authors give instead seems to suggest that only those states should be considered viable starting at which the system will stay within the desirable region if the control is NOT changed but stays constant, which is not a control problem at all but rather a question of dynamical forward invariance (in the cited paper Heitzig et al., 2016, this was called an (open) invariant kernel). Later they seem to use an even more trivial definition of "viable" by apparently identifying a viable state simply with one in which a given temperature threshold is not crossed. In the caption of Fig. 1, they suggest that by using the right policy, the state may become viable again. In viability theory, a viable state can be reached from a non-viable one only if the latter is outside the desirable region, and this reachability question is normally analysed with the concept of capture basin which is however not even mentioned here.*

**Reply:**

The reviewer is correct that we do not deal with the general control problems which are central in viability theory. Accordingly, we have rewritten section 2.1 of the paper to clarify that we use only the concept of a viable state.

*2. The study does not seem to be about a "point of no return" at all but rather about what one might call a "point of sure exceedance". For one thing, in all presented scenarios it is always possible to return to a 280 ppm world and thus to a below-threshold temperature world, so being able to return is not a question. The question is rather, at what point in time t does a (temporary) exceedance of*

*a given temperature threshold become highly likely if no mitigation has happened before t? While this is an important question, it will typically be the case that, due to the inertia of the system, this point in time is somewhat earlier than the point in time at which the threshold will actually be exceeded. So, at the critical point in time, the system is still in the desirable region but is deemed to leave it later. But this means that calling this critical point in time a "point of no RETURN" quite confusing since at this point the system has not even LEFT the desirable region.*

**Reply:**

In our paper, the point of no return is connected to a set of allowed mitigation options $F_\lambda$, which in general does not include the possibility to return to a 280 ppm world (the notion of an 'extreme mitigation scenario'). Hence, we keep the original terminology in section 2.3 but have clarified this in section 2 and in section 5.

*3. Since the authors use a one-dimensional model in section 3 and cite both classical viability theory and the framework in Heitzig et al. 2015, which are readily applicable to 1d-problems, it would be very interesting to see a comparison of their results with what these two established frameworks would yield. In section 4, they use a model which I cannot judge due to missing experience, but which seems to be still one-dimensional (dealing again with global mean temperature alone) rather than high-dimensional as claimed in the introduction. If the authors mean to say that the high dimensionality arises because of the stochastic nature of the models, I would agree in general if they really attempted to model arbitrary nonparameterized probability distributions. But on line 273 they say they assume it to be a normal distribution, so the actual model dimension is still only two (mean and variance of the distribution suffice to describe the state) and the established frameworks would again be readily applicable.*

**Reply:**

The PLASIM model has a high-dimensional state vector (the model has about $10^5$ degrees of freedom as is now mentioned in the revised section 4.1) and by linear response theory we only estimate the pdf of one component of the state vector (mean and variance).

While we agree that such as comparison would be interesting (and a topic of a follow-up paper), it is outside the scope of this paper. We discussed this with the editor and he agreed with us on this issue.

[revised manuscript text omitted]